# VeriThoughts: Enabling Automated Verilog Code Generation using Reasoning and Formal Verification

**Patrick Yubeaton**    **Andre Nakkab**    **Weihua Xiao**    **Luca Collini**

**Ramesh Karri**    **Chinmay Hegde**    **Siddharth Garg**

NYU Tandon School of Engineering
Brooklyn, NY

## Abstract

This paper introduces VeriThoughts, a novel dataset designed for reasoning-based Verilog code generation. We establish a new benchmark framework grounded in formal verification methods to evaluate the quality and correctness of generated hardware descriptions. Additionally, we present a suite of specialized small-scale models optimized specifically for Verilog generation. Our work addresses the growing need for automated hardware design tools that can produce verifiably correct implementations from high-level specifications, potentially accelerating the hardware development process while maintaining rigorous correctness guarantees. Our code and data are available at this URL.

## 1 Introduction

Large language models (LLMs) have demonstrated impressive capabilities in generating software code (such as in programming languages like Python, Java, and C++) from natural language prompts[1]. This has resulted in widespread adoption of these tools in real-world software engineering workflows, improving developer productivity by up to $55.8\%$[2]. However, notwithstanding a growing body of research, code generation for hardware design languages (HDL) has proven to be much more challenging. Verilog, one of the most commonly used HDLs, allows designers to specify the function of a chip at a high level, leaving its conversion to a manufacturable circuit to automated tools. Even so, HDL programming is notoriously tedious and time-consuming[3]. LLM-based automatic Verilog code generation would therefore significantly improve chip design productivity.

A key challenge in automating HDL code generation is the scarcity of HDL codes on the web compared to languages like Python[4]. Code generation models trained on open-source code repositories tend to struggle on Verilog problems[5]. Recent efforts have sought to build both training and evaluation datasets to remedy this issue[5,6,4,7,8]. An early effort, VeriGen[4], scraped more than 108K Verilog files from GitHub, which were then used to fine-tune open-source LLMs using self-supervised next token prediction. A small evaluation dataset of 17 natural language prompts, along with test inputs and desired outputs was proposed to benchmark the accuracy of fine-tuned LLMs. Larger training sets of HDL/Verilog code[7] and smaller evaluation datasets of prompts and corresponding test inputs/outputs[9,8] have been released.

Training sets containing Verilog code alone[6,10] can only be used to finetune LLMs with self-supervised next token prediction, but not for subsequent supervised fine-tuning (SFT) steps that are important for performance[11]. For SFT, pairs of natural language prompts (or Verilog coding *questions*) and corresponding Verilog implementations are needed. Manually writing descriptive prompts/questions for thousands of Verilog modules is infeasible, and automated prompt/question generation methods have been plagued by hallucinations[12–14] The recent success of reasoning models like DeepSeek-R1[15] and Qwen3[16] suggests that SFT on chain-of-thought (CoT) traces

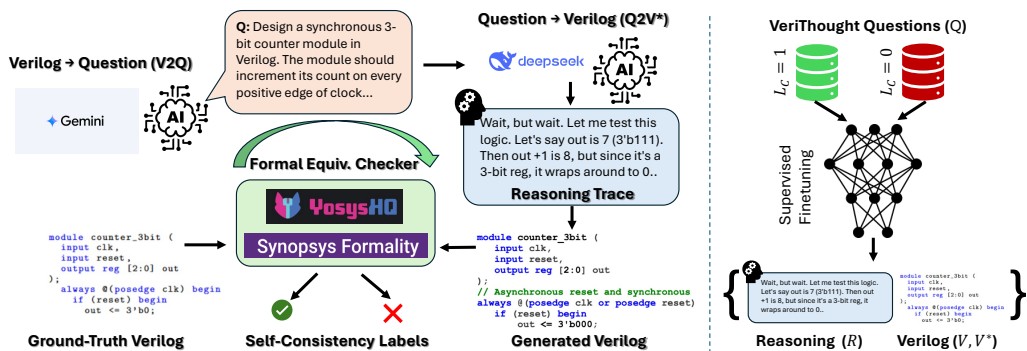

Figure 1: Generation of the VeriThoughts dataset involves four steps. Starting with a repository of ground-truth Verilog $V$, we ask a frontier model to pose a question $Q$ for which $V$ is a valid response. Then we pose $Q$ to a second frontier reasoning model, obtaining response $R$ and generated Verilog $V^*$. Finally, a formal equivalence checker $E$ returns a self-consistency label $L_C \leftarrow E(V = V^*)$. A data point in the VeriThoughts data sample is a tuple $\{V, Q, R, V^*, L_c\}$. VeriThoughts questions are used for supervised fine-tuning of SoTA LLMs with reasoning and Verilog code as targets.

can improve performance, but CoT traces for Verilog code generation are not publicly available. Moreover, evaluation datasets require mechanisms to check the correctness of generated code—this has typically been done using test inputs/outputs in both the software and hardware communities. But test inputs/outputs may not be readily available for code scraped from open repositories, and LLM generated tests are known to be buggy[17].

In this paper, we present **VeriThoughts**, the first large-scale dataset containing Verilog code with (a) paired prompts/questions, (b) prompt/question quality labels, and (c) reasoning traces for over 20,000 Verilog modules. As an ancillary outcome from our training datset, we also curate the VeriThoughts validation set with 250 questions sampled randomly from VeriThoughts, golden Verilog responses, and a *formal verification* based evaluation; this style of evaluation is a departure from the (weaker) testbench simulation evaluations used in the literature thus far.

Our data generation process shown in Figure 1 starts with MetRex[10], one of the largest datasets of "synthesizable" Verilog (i.e., Verilog that can be mapped to a hardware implementation) collated from many sources. Like VeriGen, Metrex has Verilog code only. Using a frontier Gemini model, we first produce a question ($Q$) to which a MetRex code sample ($V$) is a correct response. A second model, Deepseek-R1-670B, is used to generate a reasoning trace ($R$) and candidate Verilog implementation ($V$) responsive to the question ($Q$). However, the question and/or the generated Verilog code might be incorrect due to hallucinations, resulting in a noisy dataset. To address this concern, we propose a formal verification-based "self-consistency" check that compares ground-truth Verilog with generated Verilog; a mismatch indicates that one or more among $Q$, $R$ or $V^*$ are incorrect. Manual analysis of a random subset of training data confirms value of these labels. We demonstrate the value of our dataset by enabling answers to interesting RQs about the usefulness of reasoning traces and self-consistency labels, and by training SoTA open-source models for Verilog generation.

## 2   Background & Related Work

LLMs have demonstrated strong capabilities in code generation for programming languages such as C and Python[18–24]. This success is largely attributed to the availability of extensive training datasets comprising source code in one or multiple programming languages. These datasets can span several hundreds of gigabytes, enabling LLMs to learn syntactic patterns, semantic structures, and common usage conventions. Inputs to the models include natural language instructions, comments, partial code snippets, or combinations thereof.

Researchers explored fine-tuning open-source LLMs for HDL generation[25,12,26]. To assess performance, benchmarks such as Verilog Eval[13] have been introduced to evaluate Verilog generation

Table 1: Comparison across existing large-scale Verilog datasets. Factors include synthesizability checking, natural language questions/descriptions, availability of reasoning traces, and self-consistency checking.

| Dataset | Design Count | Synth. Verilog | Nat. Lang. Qns. | Reasoning | Self-Cons. Check |
|---|---|---|---|---|---|
| Verilog GitHub (VeriGen) | 108 971 | ✗ | ✗ | ✗ | ✗ |
| MG-Verilog | 11 144 | ✓ | ✗ | ✗ | ✗ |
| Resyn (RTL-Coder) | 26 532 | ✗ | ✓ | ✗ | ✗ |
| MetRex | 25 868 | ✓ | ✓ | ✗ | ✗ |
| **VeriThoughts (train)** | 20 173 | ✓ | ✓ | ✓ | ✓ |
| **VeriThoughts (test)** | 291 | ✓ | ✓ | ✓ | ✓ |

quality. Despite these efforts, even fine-tuned models often lag behind leading closed-source, proprietary LLMs such as Claude Sonnet or the GPT series in output quality and reliability.

In order to enable training of domain-specific Verilog generation models, a number of relevant datasets have been created. The Verilog GitHub dataset[25] includes approximately 108,000 Verilog files scraped from GitHub, and has been used to train models such as VeriGen and CL-Verilog. Building on this dataset, MG-Verilog introduced a multi-grained version of the Verilog GitHub dataset with corrected syntax. The RTL-Coder Resyn dataset utilizes LLMs to create a number of machine-generated Verilog samples[12]. Finally, the MetRex dataset, which we build upon here, contains over 25,000 Verilog files which are synthesizable via Yosys, and includes natural language descriptions of all relevant post-synthesis metrics. Table 1 contains comparisons across existing large-scale Verilog datasets, including the number of Verilog designs they include, whether they have been checked for synthesizability, whether they include natural language descriptions, whether they include a reasoning trace related to each design, and whether they include self-consistency checks.

## 3 VeriThoughts Dataset

We now introduce **VeriThoughts**, a unique, large-scale, **formally-verified Verilog reasoning** dataset. VeriThoughts is comprised of 20K samples of Verilog RTL code, each paired with a prompt describing the code, newly generated Verilog code from the prompt, the reasoning traces used to generate the new Verilog code, and a label indicating whether the generated Verilog and original Verilog are functionally equivalent. The original RTL code, $V$, is sourced from the MetRex dataset[10]. We then generate prompts, $Q$, for each Verilog entry with the Gemini-2.0-Flash-Thinking-Experimental model. These prompts are then given to Deepseek-R1 which generates Verilog code $V^*$ and reasoning trace $R$. Finally, we perform formal verification $E$ of the original and generated Verilog code with the Yosys framework to obtain a self-consistency label $L_C \leftarrow E(V = V^*)$. An overview of this framework can be seen in Figure 1.

### 3.1 MetRex: Original Verilog Dataset

The MetRex dataset, released under the BSD 3-Clause license[10], contains 25.8K Verilog designs taken from publicly available sources. These include machine generated designs such as the RTL-Coder dataset[7] and human created designs such as the VeriGen dataset[4]. The dataset creators took scraped Verilog designs from the web and cleaned the data to ensure that only synthesizable designs remained. Note that some of the RTL included in this dataset[7] include LLM-generated descriptions of their function. However, the quality of these annotations is unverified and for our purposes, we only use the RTL, and discard all existing generated text descriptions. We filter the dataset to around 25K entries that all have a character length of less than 10,000 characters.

### 3.2 Generating Prompts for Unlabeled Verilog Code

Given a Verilog sample $V$, we need to annotate this sample with a question $Q$ which accurately defines $V$. If we are to use LLMs for this annotation task, a naive prompt can lead to a litany of problems in the generated question. For example, a simple prompt such as "Create a question whose

answer is the following Verilog Code:" will often generate questions that do not explicitly state the variable names for the inputs/outputs of the module. This makes it difficult to perform formal verification on the Verilog code generated from this question. Therefore, we specify in the prompt that the output should include the name of the module and its input/output variables.

Another concern when using LLM-generated labels is the risk of the LLM "spelling out" the answer to the problem when creating a question. We sometimes find the LLM-generated question to include implementation details within the question that reveal significant portions of the original Verilog code. Therefore, we constrain the generated question by asking the LLM to generate a question that leaves room for the reader to think about the question. Finally, we find sometimes that the annotation model generates parts of a question, or earlier versions of a question, in the reasoning traces. This makes it difficult to parse out the question in a programmatic fashion. Thus, we prompt the LLM to encase the final question in a set of words that can be parsed automatically. These requirements inform our final question generating prompt, shown in Figure 2.

---

**Question Generating Prompt:**

```
Write a question whose answer is the following Verilog code.  Do not make the
question so detailed that someone can effectively copy the code straight from
the question.  The question needs to leave room for the person reading it
to need to think about the answer.  Make sure to state the interface in your
question.  You should specify the inputs and outputs and make sure they have the
same names as in the original code.  In addition, include the exact name of the
module.  Please do the same for all modules present in the Verilog code I give
you.  The beginning of your final question should start with QUESTION BEGIN and
the end of your question should end with QUESTION END.
```

---

Figure 2: Prompt used to generate a question $Q$ consistent with ground-truth Verilog ($V$).

This question generating prompt along with $V$, is used to query the Gemini annotator model to generate a $Q$ whose answer is $V$. A specific example of $V$ and $Q$ can be seen in 8.

Notwithstanding incorrect questions, other work on software code generation has found that even inconsistent question-answer pairs in a training dataset can help improve code generation accuracy[27]. To help resolve the issue in the context of Verilog code generation, VeriThoughts also includes additional labels that attempt to capture the impact of consistency between pairs of $(V, Q)$. Unfortunately, manual annotations are not feasible for a dataset of our size—instead, we employ *self-consistency check*, described below, as a proxy for question answer consistency.

### 3.3 Generating Reasoning Traces

We choose the DeepSeek-R1 model as our Verilog code generation model. It is notable for being one of the best performing open source models on coding problems while also being a reasoning model. Thus, when generating Verilog code from our generated question, we can extract the reasoning traces, $R$, produced from this inference. We store $R$ in the VeriThoughts dataset to be analyzed and used in later model training. We follow the recommended generation guidelines for DeepSeek-R1 by using a temperature setting of 0.6, a top_p of 0.95, and a maximum generation length of 8192 tokens. We append the following statement to every question: "Make sure your input and output interface has the same names as described in the question. \nPlease start your Verilog code with CODE BEGIN and end with CODE END."+ "\n<think>\n". We instruct DeepSeek-R1 to stay consistent with module interface because it sometimes changes the module or variable names. In addition, we add code parsing bookends to the prompt (similar to those present during question generation). Finally, following best practices for DeepSeek we append a <think> token to produce reasoning traces.

### 3.4 Verifying Question Quality using Self Consistency

Given a $(V, Q)$ pair, it is difficult to *a priori* ascertain the quality of the question $Q$. However, we leverage the following key insight: if one were to use $Q$ to generate *new* Verilog code $V^*$, then we could compare the functional equivalence of this newly generated code and the original Verilog as a *proxy* for the question quality. If $(V, V^*)$ are equivalent, we can be highly confident that the question accurately describes the golden Verilog. However, if $(V, V^*)$ are not functionally equivalent, there

is a possibility of a mismatch between $(V, Q)$, $(Q, V^*)$, or both. Therefore, we use the generated questions to perform this "self-consistency" check as a proxy for the quality of $Q$.

Given the newly generated $V^*$, we can now perform formal verification to see if the generated code is functionally equivalent to the original (or "golden") Verilog. Formal verification tools mathematically prove the equivalence of two circuits for all possible input combinations. This is much more powerful than standard practice in LLM evaluations, which lean on human-designed unit tests, or LLM-as-judge tests. In addition to being a stronger equivalence check than human-designed test cases, using a formal verification tool takes away the necessity of manual (or LLM-enabled) generation of test cases. We perform formal verification with the Yosys software using the script shown in A.

The script begins by loading in the two sets of Verilog: golden and generated. Afterwards, the third line prepares the Verilog code for synthesis by performing various optimizations. The fourth line converts clocked flip flops into combinational logic which is necessary for performing verification on sequential circuits. The fifth line creates a specific circuit (miter) used for equivalence checking. The final line runs a Boolean Satisfiability (SAT) solver to see if the two circuits are equivalent. We take the output of $L_C \leftarrow E(V = V^*)$ and add it to the VeriThoughts dataset with the remaining entries in the tuple $\{V, Q, R, V^*, L_c\}$. In addition, there are cases where $V$ contains multiple modules. We do sub-module level verification by checking $E(V_s = V_s^*)$ for each sub-module pair $(V_s, V_s^*) \in V$. If all sub-modules are functionally equivalent, then $L_c \leftarrow 1$, otherwise $L_c \leftarrow 0$.

### 3.5 VeriThoughts: Statistics and Analysis

We explore various statistics about the Verilog dataset we have generated. In Figure 3 we compare the self-consistent ($L_c = 1$) and inconsistent ($L_c = 0$) datasets. We examine the number of lines in the Verilog ground truth ($V$), the number of modules in ($V$), the number of sequential code samples in each dataset, and the number of characters in the reasoning traces ($R$). We see that the distributions are quite different between the two datasets. The number of lines is centered around 24 lines of code for the consistent dataset, but is centered around 50 lines of code. We see a similar trend in the number of characters present in each dataset's reasoning traces.

These two factors combined suggest that the Verilog present in the inconsistent dataset is more difficult or complex than the Verilog present in the self-consistent dataset. An additional point of support for this can be seen in the right most graph which compares the number of combinational and sequential circuits in each dataset. We see an increased number of sequential circuits in the inconsistent dataset which matches with the real world expectation that sequential circuits are more complex than combinational circuits.

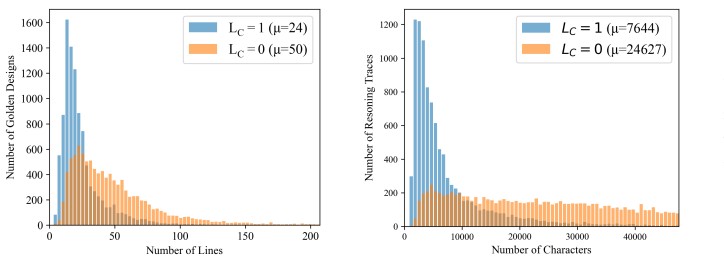

Figure 3: These figures explore various statistics when comparing the self-consistent and inconsistent subset of VeriThoughts. The left figure compares the number of lines of code in ground truth Verilog samples. The middle figure compares the number of characters in a reasoning trace. The right figure looks at the number of sequential modules present in each dataset.

### 3.6 Manual Analysis of VeriThoughts Subset

To validate the effectiveness of our formal-verification-based dataset-generation framework, we randomly sampled 100 data points from our VeriThoughts dataset, which are further evaluated manually. Each data point contains a tuple $(V, V^*, Q)$. The 100 samples are comprised of 50 data points that passed the self consistency test, and 50 data points that fail to pass the self consistency test. For every sampled data point, a human expert reviewer evaluated whether the $Q$ *accurately*

summarizes $V$, and whether $V^*$ *accurately* implements $Q$. Combining the answers to these two questions yields four possible evaluation cases:

- Prompt error: the prompt does not summarize the golden Verilog accurately, but the generated Verilog accurately follows the prompt.
- Code-gen error: the prompt is accurate, yet the generated Verilog deviates from it.
- Both errors: the prompt inaccurately summarizes the golden Verilog, and the generated Verilog inaccurately implements the prompt.
- No error: no mismatches are found in the two aspects.

Table 2 reports the distribution of different evaluation cases. The first row gives raw counts for each subset (matched or mismatched), and the second row gives percentages relative to that subset.

Table 2: Outcomes of our manual analysis of 100 datasamples from the VeriThoughts dataset, with 50 drawn at random from Self-consistent ($L_C = 1$) responses and 50 from Inconsistent responses ($L_C = 0$). Incorrect questions are inconsistent with ground-truth Verilog ($V$). Incorrectly generated Verilog ($V^*$) is when the question ($Q$) is correct but $V^*$ is inconsistent with $Q$.

| Dataset | Question Incorrect | Generated Verilog Incorrect | Both Incorrect |
|---|---|---|---|
| Self-Consistent (50) | 0 (0.0%) | 0 (0.0%) | 0 (0.0%) |
| Inconsistent (50) | 33 (66.0%) | 8 (16.0%) | 9 (18.0%) |

We provide three representative examples to illustrate the prompt error (Fig. 4), code-gen error (Fig. 5), and both errors (Fig. 6) cases identified above. Among the $50$ mismatched data points, most of them ($66\%$) correspond to the prompt error case: the prompt produced by the LLM cannot accurately explain the golden Verilog code. A smaller fraction ($16$ %) correspond to the code-generation error case, where the prompt is accurate but the LLM implements it inexactly. Finally, $18$ % belong to the both errors case; here a common pattern is that the generated prompt omits some implementation details of the golden Verilog, and the LLM fills in the missing details, leading to function mismatches.

## 4 Applications of VeriThoughts

The VeriThoughts dataset lets us pose interesting research questions for studying Verilog generation models that are otherwise difficult to explore systematically. Such questions include: How do reasoning traces impact the utility of a Verilog model? How important is it for a question answer pair to be consistent with each other? Do hallucinated prompts paired with hallucinated answers provide downstream utility? For the rest of this section, we will first outline the essential benchmarks and finetuning tools necessary to tackle the aforementioned questions. Then, we finetune several models using VeriThoughts. Finally, we will compare these models against other Verilog generation models.

### 4.1 Experimental Setup

**LLM Finetuning** In all subsequent experiments we perform supervised fine-tuning (SFT) of Qwen-2.5-Instruct-7B models using a learning rate of $8 * 10^{-5}$, a cosine scheduler, `bf16` format, and three epochs over our training set. This finetuning is done with Llama-Factory[28]. The full fine-tuning was performed on 8xA100 GPUs and takes roughly 2-4 hours for instruction tuned models and roughly 7-14 hours for reasoning models.

**Evaluations** We evaluate both preexisting and our trained models on two benchmarks. VerilogEval[5] has 156 problems sourced from HDLBits and is meant to test a language model's ability to generate a diverse distribution of Verilog code. We focus on the "Human" split of VerilogEval 1.0 because it has higher-quality labels and problems than the machine-generated "Machine" split. The benchmark is evaluated using human-designed test cases for every Verilog problem.

In addition to VerilogEval, we create a new benchmark sourced by randomly sampling a hold-out set of 291 questions from VeriThoughts. We call this the *VeriThoughts benchmark*, and validate it using the formal verification scheme described in Section 3. This helps avoid the strain of manually creating unit test cases (such as in VerilogEval) while also providing a stronger level of verification

Table 3: Dataset subsets derived from VeriThoughts. Subsets are defined by features of the data points such as self-consistency and the type of verilog target used for fine-tuning.

| Dataset Label | Self-Consistent | Fine-tune Type | Verilog Target |
|---|---|---|---|
| A | Yes | Reasoning | Ground Truth |
| B | Yes | Instruct | Ground Truth |
| C | No | Reasoning | Ground Truth |
| D | No | Instruct | Ground Truth |
| E | No | Reasoning | Generated |
| F | No | Instruct | Generated |

for our benchmark. We generate responses for VerilogEval with a temperature of 0.5, top_p of 0.90, and maximum generation length of 1024 tokens. This is in line with the VerilogEval Human 1.0 results[13]. For our VeriThoughts benchmark, we generate responses with a temperature of 0.6, top_p of 0.95, and maximum generation length of 16384. Temperature and top_p are in line with the suggested generation settings for DeepSeek and the maximum generation length is increased for longer problems (and associated reasoning traces) in VeriThoughts. We report standard "pass@k"[29] metrics for $k = \{1, 5, 10\}$, standard values used in the literature. Pass@k is evaluated over 20 trials.

**Dataset Subsets**   We create dataset subsets to measure the impact of different features in the dataset on downstream performance. The subset list is found in Table 3. We see that datasets A and B are the only self-consistent datasets. Of those two, we see that A is a reasoning style dataset while B is a instruction tuning dataset. Datasets C and D follow a similar pattern except they are not consistent datasets. Finally, E and F use the generated Verilog as the training target instead of the ground truth Verilog. All subsets are approximately 10K samples to ensure fairness in the dataset size.

## 4.2   Research Questions

The new features of the VeriThoughts dataset enable us to answer interesting research questions of interest to the community, which have previously not been addressed in literature.

**RQ1: Does reasoning help Verilog code generation?**   Reasoning in LLMs has been shown to help in tasks such as coding and science[30,31]; does this apply to a low-resource coding language such as Verilog? We validate this hypothesis by comparing models trained on full reasoning traces versus instruction tuning along. The results are in Table 4.

Table 4: Pass@k scores comparing Reasoning vs. Instruct models across VeriThoughts and Verilog Eval benchmarks. Diff. denotes the difference between the pass@k accuracies of the reasoning and instruct models. Bold entries are the highest scoring models for a specific pass@k. Underlined entries are the second highest scoring models for a specific pass@k.

| Model | Self-Consistent | Verilog Target | VeriThoughts | | | Verilog Eval | | |
|---|---|---|---|---|---|---|---|---|
| | | | Pass@1 | Pass@5 | Pass@10 | Pass@1 | Pass@5 | Pass@10 |
| Reasoning | Yes | Ground Truth | 75.5% | 88.9% | 92.1% | 34.6% | 47.2% | 50.7% |
| Instruct | Yes | Ground Truth | 49.0% | 69.0% | 73.7% | 21.9% | 31.6% | 34.6% |
| | | Diff. | 26.5% | 19.9% | 18.4% | 12.7% | 15.6% | 16.1% |
| Reasoning | No | Ground Truth | 51.0% | 79.3% | 86.4% | 37.2% | **52.1%** | **56.8%** |
| Instruct | No | Ground Truth | 44.9% | 65.3% | 72.4% | 19.8% | 31.2% | 35.4% |
| | | Diff. | 6.1% | 14.0% | 14.0% | 17.4% | 20.9% | 21.4% |
| Reasoning | No | Generated | **82.8%** | **94.2%** | **95.7%** | 37.3% | 50.1% | 53.0% |
| Instruct | No | Generated | 51.0% | 68.9% | 73.5% | **37.9%** | 43.2% | 44.7% |
| | | Diff. | 31.8% | 25.3% | 22.2% | -0.6% | 6.9% | 8.3% |

We see that models trained on reasoning datasets (A, C, E) outperform the corresponding models trained on instruction datasets (B, D, F) for VeriThoughts and Verilog Eval. The one exception to this trend is Datasets E and F where there is an essential tie between reasoning versus vanilla instruct

for Verilog Eval. (However, we see that for pass@5 and pass@10 that dataset E returns to being the more effective model.) These results suggest that reasoning benefits Verilog coding tasks.

**RQ2: Does question-answer consistency improve accuracy?** For a traditional code generation dataset, it is extremely hard to guarantee the consistency of a question-answer pair in the absence of human expert evaluation. However, VeriThoughts starts with synthesizable Verilog, which helps us generate consistency labels for ground truth Verilog $V$ and questions $Q$. Therefore, we can measure how consistency impacts the downstream coding ability of Verilog fine-tuned models. We

Table 5: Pass@k scores comparing Self-Consistent vs. Inconsistent examples for Reasoning and Instruct models across VeriThoughts and Verilog Eval. Diff. denotes the difference between the pass@k accuracies of the pass and fail models. Bold entries are the highest scoring models for a specific pass@k. Underlined entries are the second highest scoring models for a specific pass@k.

| Model | Self-Consistent | Verilog Target | VeriThoughts | | | Verilog Eval | | |
|---|---|---|---|---|---|---|---|---|
| | | | Pass@1 | Pass@5 | Pass@10 | Pass@1 | Pass@5 | Pass@10 |
| Reasoning | Yes | Ground Truth | **75.5%** | **88.9%** | **92.1%** | 34.6% | 47.2% | 50.7% |
| Reasoning | No | Ground Truth | 51.0% | 79.3% | 86.4% | **37.2%** | **52.1%** | **56.8%** |
| | | Diff. | 24.5% | 9.6% | 5.7% | -2.6% | -4.9% | -6.1% |
| Instruct | Yes | Ground Truth | 49.0% | 69.0% | 73.7% | 21.9% | 31.6% | 34.6% |
| Instruct | No | Ground Truth | 44.9% | 65.3% | 72.4% | 19.8% | 31.2% | 35.4% |
| | | Diff. | 4.1% | 3.7% | 1.3% | 2.1% | 0.4% | -0.8% |

see in Table 5 that the majority of the data points benefit from a self-consistent question answer pair. However, we do notice on VerilogEval that the reasoning model does not benefit from this consistency at all pass@k levels. Therefore, it appears that consistency is not the only factor affecting the downstream utility. One additional factor to consider is the contents of the datasets. We saw in Section 3.5 that the inconsistent dataset is comprised of Verilog modules with a higher number of lines of code and a longer reasoning trace than the consistent dataset. This suggests the two subsets have slightly different task distributions, and can explain the discrepancy in Table 5.

**RQ3: Do hallucinated prompts paired with hallucinated answers provide downstream utility?** We have shown that on average models benefit from being trained on reasoning datasets that are self-consistent. However, self-consistency has been explored solely from the perspective of ground truth $V$ and $Q$. Our human evaluations in Section 3.6 show that the vast majority of the mismatches between $(V, V^*)$ are caused by a mismatch in $(V, Q)$ rather than a mismatch in $(Q, V^*)$. Therefore, we test the impact of a training set built on $(Q, V^*)$ instead of $(Q, V)$. The results are shown in Table 6. We see that models trained on the inconsistent subset $(Q, V^*)$ outperform models trained on

Table 6: Pass@k scores comparing Self-consistent vs. Inconsistent (Generated) examples for Reasoning and Instruct models across VeriThoughts and Verilog Eval. Diff denotes the difference between the pass@k accuracies of the pass and fail models. Bold entries are the highest scoring models for a specific pass@k. Underlined entries are the second highest scoring models for a specific pass@k.

| Model | Self-Consistent | Verilog Target | VeriThoughts | | | Verilog Eval | | |
|---|---|---|---|---|---|---|---|---|
| | | | Pass@1 | Pass@5 | Pass@10 | Pass@1 | Pass@5 | Pass@10 |
| Reasoning | Yes | Ground Truth | 75.5% | 88.9% | 92.1% | 34.6% | 47.2% | 50.7% |
| Reasoning | No | Generated | **82.8%** | **94.2%** | **95.7%** | 37.3% | **50.1%** | **53.0%** |
| | | Diff. | -7.3% | -5.3% | -3.6% | -2.7% | -2.9% | -2.3% |
| Instruct | Yes | Ground Truth | 49.0% | 69.0% | 73.7% | 21.9% | 31.6% | 34.6% |
| Instruct | No | Generated | 51.0% | 68.9% | 73.5% | **37.9%** | 43.2% | 44.7% |
| | | Diff. | -2.0% | 0.1% | 0.2% | -16.0% | -11.6% | -10.1% |

the consistent subset for nearly all evaluations. This suggests that there may be value in hallucination question-answer pairs, but why? In Section 3.5 we saw that the inconsistent dataset generally had more lines of code, more sequential modules, and longer reasoning traces. It is often the case that sequential Verilog code is more complex than combinational Verilog code. In addition, it can be argued that code with more lines or code that induces longer reasoning traces may also be more difficult on average. The final observation can be made from Section 3.6 where we saw that the vast

majority of inconsistent data had consistent $(Q, V^*)$, but inconsistent $(Q, V)$. Therefore, one possible explanation for this observation is that the hallucinated prompt accurately describes the generated Verilog. In addition, if the "inconsistent" question answer pairs are more difficult on average than the "consistent" question answer pairs, it would explain this downstream behavior. Although we cannot concretely determine the reasons for this behavior, we are only able to explore this question with such detail because of the modular nature of VeriThoughts.

### 4.3 A new state of the art on VerilogEval

Given the quality and scale of the VeriThoughts dataset, our goal is to leverage it to create new state-of-the-art Verilog generation models. The results for various open and closed source models on the VeriThoughts and VerilogEval benchmarks are shown in Table 8.

Table 7: Pass@k scores for closed- and open-source models on VeriThoughts and Verilog Eval. Closed source models first, followed by open source models. Bold entries are the highest scoring models for a pass@k. Underlined entries are the second highest scoring models for a pass@k. Missing evaluations are due to compute constraints and will be updated in the supplementary material.

| Model | Model Type | VeriThoughts | | | Verilog Eval | | |
|---|---|---|---|---|---|---|---|
| | | Pass@1 | Pass@5 | Pass@10 | Pass@1 | Pass@5 | Pass@10 |
| **Closed Source** | | | | | | | |
| ChatGPT-o3 | Reasoning | **92.6%** | 96.7% | 97.6% | **74.4%** | **84.7%** | **86.8%** |
| Gemini-2.5-Flash-Preview-04-17 | Reasoning | 88.4% | **97.3%** | **98.3%** | 54.9% | 70.7% | 76.0% |
| **Open Source** | | | | | | | |
| Qwen3-14B | Reasoning | **87.4%** | **96.8%** | **98.1%** | 28.1% | 45.6% | 51.0% |
| VeriThoughts-14B (Qwen2.5 Base) | Reasoning | 78.5% | 90.0% | 92.1% | **43.7%** | 52.2% | 55.14% |
| DeepSeek-R1-Distill-Qwen-14B | Reasoning | 46.2% | 81.4% | 89.1% | 38.7% | **62.1%** | **69.0%** |
| Qwen2.5-7B | Instruct | 40.6% | 68.9% | 76.4% | 25.6% | 38.9% | 42.6% |
| Qwen2.5-14B | Instruct | 36.9% | 75.0% | 84.7% | 30.7% | 50.2% | 56.8% |
| CodeLlama-13B | Instruct | 27.1% | 60.9% | 72.6% | 20.7% | 35.8% | 41.7% |
| CodeLlama-13B-Python | Base | 12.4% | 40.9% | 56.6% | 24.1% | 41.3% | 47.9% |
| CL-Verilog-7B | Base | 10.2% | 36.9% | 53.6% | 21.7% | 36.4% | 42.7% |
| CL-Verilog-13B | Base | 5.0% | 19.6% | 30.9% | 26.0% | 42.1% | 47.7% |
| Llama 3.1-8B | Instruct | 6.8% | 25.9% | 40.5% | 18.9% | 33.2% | 37.4% |

We train a 14B reasoning model using dataset fold A, with Qwen-2.5-Instruct-14B as the base. We see that our model performs very competitively among open source models. On the standard VerilogEval benchmark (at pass@1) our model has the highest accuracy, and is close behind DeepSeek-R1-Distill-Qwen-14B for pass@5 and pass@10. Moreover, our model shines on our new VeriThoughts benchmark, only losing out to the recently released Qwen3-14B model. These results show that even with a small fine-tuning set (with ∼10K samples) we are able to create a powerful Verilog reasoning model that can compete with other reasoning and instruction-tuned models.

## 5 Discussion

In this paper, we introduced VeriThoughts, the first large scale formally verified reasoning dataset for Verilog. We use VeriThoughts to tackle previously difficult-to-approach research questions such as the value of hallucinated prompts. We then use VeriThoughts as a training set to fine-tune a state-of-the-art (Open Source) Verilog generation model with only 10K training samples.

We consider this work to be just the start for generating data via formal verification. We believe that adding new data sources outside of MetRex will lead to more diverse Verilog code generations. In addition, we would like to apply our formal verification benchmarking methodology to existing Verilog benchmarks such as Verilog Eval to see how those benchmarks are impacted. We hope that this work will increase interest in formal verification and reasoning for dataset generation.

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

Figure 4: Example of prompt error case.

# A  Appendix

**Yosys Script:**
read_verilog verilog_truth.v
read_verilog verilog_gen.v
prep; proc; opt; memory;
clk2fflogic;
miter -equiv -flatten new_module_name original_module_name miter
sat -seq 50 -verify -prove trigger 0 -show-all -show-inputs -show-outputs -set-init-zero miter

[Golden Verilog]
```verilog
module top_module(
    input [3:0] a,
    input [3:0] b,
    input select,       // Select input to
choose between different outputs
    output [3:0] out_xor_bitwise,
    output out_xor_logical,
    output [7:0] out_not
);
    // Generate the inverse of the input
numbers
    wire [3:0] a_inv = ~a;
    wire [3:0] b_inv = ~b;

    // Generate the bitwise-XOR output
    wire [3:0] xor_bitwise = a ^ b;

    // Generate the logical-XOR output
    wire xor_logical = !(a & b) && (a | b);

    // Generate the output based on the
select input
    assign out_xor_bitwise = select ?
xor_bitwise : 4'b0;
    assign out_xor_logical = select ?
xor_logical : 1'b0;
    assign out_not = select ? {b_inv, a_inv}
: 8'b0;
endmodule
```

[Generated Verilog]
```verilog
module top_module (
    input [3:0] a,
    input [3:0] b,
    input select,
    output [3:0] out_xor_bitwise,
    output out_xor_logical,
    output [7:0] out_not
);

    assign out_xor_bitwise = select ? (a ^
b) : 4'd0;
    assign out_xor_logical = select ? ((|a
|| |b) && !(|a && |b)) : 1'b0;
    assign out_not = select ? {~b, ~a} :
8'd0;

endmodule
```

[Prompt]

Write a Verilog module named `top_module` with the following interface:
`input [3:0] a, input [3:0] b, input select, output [3:0] out_xor_bitwise,`
`output out_xor_logical, output [7:0] out_not`

This module should implement the following logic based on the `select` input:

*   When `select` is high (1):
    *   `out_xor_bitwise` should be the bitwise XOR of inputs `a` and `b`.
    *   `out_xor_logical` should be the logical XOR of inputs `a` and `b`. Implement this logical XOR using only basic logical
operators (AND, OR, NOT).
    *   `out_not` should be an 8-bit value formed by concatenating the bitwise inverse of `b` with the bitwise inverse of `a` (i.e.,
`{~b, ~a}`).
*   When `select` is low (0):
    *   All outputs (`out_xor_bitwise`, `out_xor_logical`, and `out_not`) should be driven to their respective zero
values (0 for 1-bit, all zeros for wider outputs).

Your implementation should use continuous assignments.

Figure 5: Example of code-gen error case.

Table 8: Pass@k scores for closed- and open-source models on Verilog Eval. Closed source models first, followed by open source models.

| Model | Model Type Pass@1 | Verilog Eval | | |
|---|---|---|---|---|
| | | Pass@5 | Pass@10 | |
| Claude 3.7 Sonnet | Reasoning | 66.3% | 75.8% | 78.5% |
| GPT-4o | Instruct (MoE) | 56.3% | 66.7% | 69.0% |
| GPT-4 | Instruct (MoE) | 43.5% | 55.8% | 58.9% |
| VeriThoughts-32B (Qwen2.5 Base) | Reasoning | 52.0% | 61.6% | 63.6% |
| VeriThoughts-32B (Qwen3 Base) | Reasoning | 49.5% | 59.2% | 61.9% |
| VeriThoughts-14B (Qwen3 Base) | Reasoning | 38.7% | 51.8% | 55.6% |
| RTLCoder-v1.1-7B | Instruct | 34.6% | 43.4% | 45.5% |
| RTLCoder-DeepSeek-v1.1-7B | Instruct | 39.7% | 49.3% | 51.9% |

**[Golden Verilog]**

```verilog
module shift_register(input clk, stb, di,
output reg do);
    localparam integer DIN_N = 256;
    localparam integer DOUT_N = 256;

    reg [DIN_N-1:0] din;
    reg [DIN_N-1:0] din_shr;
    reg [DOUT_N-1:0] dout;
    reg [DOUT_N-1:0] dout_shr;

    always @(posedge clk) begin
        din_shr <= {din_shr, di};
        dout_shr <= {dout_shr,
din_shr[DIN_N-1]};
        if (stb) begin
            din <= din_shr;
            dout <= dout_shr;
        end
    end

    always @(posedge clk) begin
        do <= dout_shr[DOUT_N-1];  //
Corrected the assignment of do to
dout_shr[DOUT_N-1]
    end

endmodule
```

**[Generated Verilog]**

```verilog
module shift_register #(
    parameter integer DIN_N = 1,
    parameter integer DOUT_N = 1
) (
    input wire clk, stb, di, output wire do
);
reg [DIN_N-1:0] first_stage;
reg [DOUT_N-1:0] second_stage;
reg [DIN_N-1:0] captured_din;
reg [DOUT_N-1:0] captured_dout;
// First stage shift logic
always @(posedge clk) begin
    first_stage <= (DIN_N > 1) ?
{first_stage[DIN_N-2:0], di} : di;
end
// Second stage shift logic
always @(posedge clk) begin
    second_stage <= (DOUT_N > 1) ?
{second_stage[DOUT_N-2:0],
first_stage[DIN_N-1]} : first_stage[DIN_N-
1];
end
// Capture on stb
always @(posedge clk) begin
    if (stb) begin
        captured_din <= first_stage;
        captured_dout <= second_stage;
    end
end
assign do = (DOUT_N >= 1) ?
second_stage[DOUT_N-1] : 1'b0;
endmodule
```

**[Prompt]**

Write a Verilog module named `shift_register` with the following interface:
Inputs: `clk`, `stb`, `di`
Output: `do`

The module should implement a two-stage serial processing pipeline.
On each rising edge of the `clk` signal:
1. A new bit `di` is shifted into the first stage.
2. The bit that has propagated fully through the first stage is shifted into the second stage.

The capacities of the first and second stages should be configurable using integer parameters named `DIN_N` and `DOUT_N`, respectively.

The output `do` should continuously provide the bit that has propagated fully through the second stage.

Additionally, when the `stb` signal is asserted high on a clock edge, the current contents of both the first and second stage shift registers should be captured and stored in separate internal registers.

Figure 6: Example of both errors case.

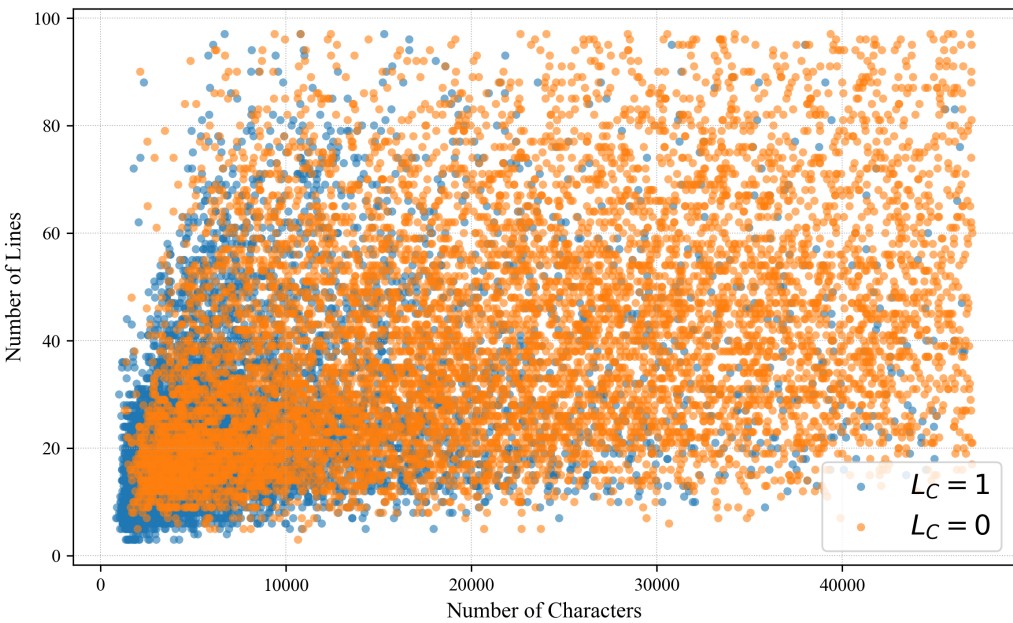

Figure 7: Number of lines of golden Verilog designs vs number of characters of reasoning traces.

| Ground-Truth Verilog, V | Question, Q | Generated Verilog, V* |
|---|---|---|
| ```verilog
module counter_3bit (
input clk,
input reset,
output reg [2:0] out
);
always @(posedge clk)
begin
    if (reset) begin
        out <= 3'b0;
    end else begin
        out <= out + 1;
    end
end
endmodule
``` | Design a synchronous 3-bit counter module in Verilog. The module should increment its count on each positive edge of the clock input (`clk`). The counter should wrap around to 0 after reaching its maximum value. Implement a synchronous reset (`reset`) that sets the counter output to 0 when asserted high. Your module must be named `counter\_3bit`. The interface should be as follows: Inputs: - `clk`: Clock signal. - `reset`: Asynchronous reset signal (active high). Output: - `out`: A 3-bit output, indexed from [2:0], representing the current count. | ```verilog
module counter_3bit (
input clk,input reset,
output reg [2:0] out
);
always @(posedge clock
    and posedge reset)
begin
    if (reset) begin
        out <= 3'b0;
    end else begin
        out <= out + 1;
    end
end
endmodule
``` |

Figure 8: Example of wrong question $Q$, wrongly capturing the functionality (in this case, reset behavior) and the generated verilog $V^*$ correctly implementing the functionality in $Q$.

| Ground-Truth Verilog, V | Question, Q | Generated Verilog, V* |
|---|---|---|
| ```verilog
module top_module(
input [3:0] a,
input [3:0] b,
input select, // Select input t
o choose between different outp
uts
output [3:0] out_xor_bitwise,
output out_xor_logical,
output [7:0] out_not);
// Generate the inverse of the
input numbers
wire [3:0] a_inv = ~a;
wire [3:0] b_inv = ~b;
// Generate the bitwise-XOR out
put
wire [3:0] xor_bitwise = a ^ b;
// Generate the logical-XOR out
put
wire xor_logical = !(a & b) &&
(a | b);
// Generate the output based on
the select input
assign out_xor_bitwise = select
? xor_bitwise : 4'b0;
assign out_xor_logical = select
? xor_logical : 1'b0;
assign out_not = select ? {b_in
v, a_inv} : 8'b0;
endmodule
``` | When `select` is high (1):
  * `out_xor_bitwise` should be the bitwise XOR of inputs `a` and `b`.
  * `out_xor_logical` should be the logical XOR of inputs `a` and `b`. Implement this logical XOR using only basic logical operators (AND, OR, NOT).
  * `out_not` should be an 8-bit value formed by concatenating the bitwise inverse of `b` with the bitwise inverse of `a` (i.e., `{~b, ~a}`).
When `select` is low (0):
  * All outputs (`out_xor_bitwise`, `out_xor_logical`, and `out_not`) should be driven to their respective zero values (0 for 1-bit, all zeros for wider outputs). | ```verilog
module top_module (
input [3:0] a,
input [3:0] b,
input select,
output [3:0] out_xor_bitwise,
output out_xor_logical,
output [7:0] out_not
);

assign out_xor_bitwise = select
? (a ^ b) : 4'd0;
assign out_xor_logical = select
? ((|a || |b) && !(|a && |b)) :
1'b0;
assign out_not = select ? {~b,
~a} : 8'd0;

endmodule
``` |

Figure 9: Example of mismatch between the question and the generated Verilog.

