# OpenReview forum: "VeriThoughts: Enabling Automated Verilog Code Generation using Reasoning and Formal Verification"
_NeurIPS.cc/2025/Datasets_and_Benchmarks_Track — NeurIPS 2025 Datasets and Benchmarks Track poster_

### Official Review · Reviewer_bfqD · 2025-06-29

**Rating:** 4
**Confidence:** 4

**Summary:**

This paper proposes a new dataset for verilog code generation. Given the ground-truth verilog code, it first utilizes LLMs to generate question which defines this verilog code. Then, reasoning models such as DeepSeep-R1 is selected to generate the new verilog code along with the corresponding reasoning traces. Experiments show that the reasoning trace (CoT) is beneficial to the code generation quality, and it also provides more explainability.

**Additional Feedback:**

1. Could more sources of data be added to increase the diversity of the dataset?
2. Could the dataset be applied into other areas such as formal verification?

**Dataset Code Accessibility:**

Yes

**Dataset Code Comments:**

The authors provide github and huggingface projects with clear tutorials about how to access and utilize the dataset.

**Ethical Comments:**

This paper focuses on verilog generation, which has no relation with other ethics flags such as human rights and environmental impact. For the data quality and data privacy problems, it utilized MetRex which is high quality and has been properly cited.

**Ethical Considerations:**

No, there are no or only very minor ethics concerns

**Final Justification:**

I think that the correctness of "incorrect responses and incorrect reasoning traces can sometimes result in similar or better performance" needs further verification, which could not be a strong evidence to support the authors' statement. Further discussion and analysis should me added to enhance the presentation. According to other reviews, I would like to keep my original rating.

**Limitations Weaknesses:**

1. Since most LLM are general purpose, how to ensure the quality of the generated questions and reasoning traces? Is it necessary to first fine-tune the LLM on the domain-specific dataset to enhance its capacity?
2. In table 5, the reasoning model with self-consistency performs worse than that without self-consistency. Could the author explain this phenomenon?

**Strengths Contributions:**

1. It utilizes the reasoning model to generate the reasoning traces, which could be a stronger and more detailed supervision signal for SFT.
2. A approach based on self consistency is proposed to verifying the question quality.
3. Experiments show that the reasoning traces and the question-answer consistency could improve the model performance, demonstrating the effectiveness of the proposed features.
4. The reasoning trace (CoT) provides more explainability of the verilog code generation process, and it is also beneficial to the code generation quality.

---

> ### Author Rebuttal · Authors · 2025-07-31
>
> We thank the reviewer for highlighting the benefits our generated reasoning traces bring to code generation performance and explainability. We address the limitations and weaknesses brought up by the reviewer below.
>
> **How to ensure the quality of generated questions/reasoning traces that come from general purpose LLMs**: We propose the Self-Consistent label exactly for this purpose. Using a formal verification tool (such as Yosys) we test whether the generated Verilog code is functionally equivalent to the ground truth Verilog code used to generate the question. If the two sets of code are functionally equivalent, we believe the question to be high enough quality to replicate the original code. Thus, our Self-Consistent label serves as a proxy for question quality and the reasoning traces.
>
> We were not able to fine-tune the frontier LLMs (Gemini and DeepSeek-R1) due to closed source and/or resource limitations; however we believe it would be a very interesting research direction. A fine-tuned teacher model would likely allow for more Self-Consistent questions to be generated due to the increased domain knowledge of the teacher model. This in turn would lead to a larger Self-Consistent dataset which we have shown (RQ2) would likely lead to downstream benefits.
>
> **Reasoning model with Self Consistent data is worse than model with  Non Self Consistent data in Table 5**:
> As noted by the reviewer, we see that the Self-Consistent labels do not lead to accuracy gains for the reasoning model on VerilogEval. Interestingly, parallel to our work, two recent papers have also shown similar results, i.e., that incorrect responses and incorrect reasoning traces can sometimes result in similar or better performance (“Interpretable Traces, Unexpected Outcomes: Investigating the Disconnect in Trace-Based Knowledge Distillation”, OpenThoughts: Data Recipes for Reasoning Models), especially when distilling smaller reasoning models. In Section 3.5, we provide some insights into why this might be the case. Specifically, we find that the Verilog  code in the non Self-Consistent dataset had more lines of code on average than the Self-Consistent dataset, as well as longer reasoning traces on average. Non Self-Consistent data also had a greater prevalence of “sequential” (clocked) code which is typically harder to program.. These observations suggest that problems in the non-Self-Consistent part of the dataset were more challenging than those in the Self-Consistent partition, thus limiting the performance of the models trained only on Self-Consistent data. Although this hypothesis needs further evidence, it illustrates the potential value of our dataset and data generation pipeline to furthering research in this direction.
>
> **Could more data be added**: Yes, we believe that adding additional sources of data to the dataset would lead to a greater diversity of Verilog problem “types” in the dataset and would likely lead to improved downstream performance. A nice feature of our data generation pipeline is that it can be applied to any repository of raw Verilog code, including other open datasets like opencores or leveraged within hardware companies on their internal repositories.
>
> **Use in formal verification**: Yes , the reviewer makes an excellent point. The data we have gathered also creates a new dataset for formal verification since we have paired instances of Verilog code and labels suggesting whether these are equivalent or not. This is an interesting area for future work.

---

> > ### Comment · Reviewer_bfqD · 2025-08-03
> >
> > Thank you for your response. I think that the correctness of "incorrect responses and incorrect reasoning traces can sometimes result in similar or better performance" needs further verification, which could not be a strong evidence to support the authors' statement. Further discussion and analysis should me added to enhance the presentation. According to other reviews, I would like to keep my original rating.

---

### Official Review · Reviewer_vynn · 2025-07-03

**Rating:** 4
**Confidence:** 3

**Summary:**

This paper propose a reasoning Verilog training and evaluation dataset containing 20k and 291 samples respectively. To ensure the reasoning and generated results correct, the paper propose Self-Consistent method to control the quality.

**Dataset Code Accessibility:**

Yes

**Ethical Considerations:**

No, there are no or only very minor ethics concerns

**Final Justification:**

Overall, most of my problems have been resolved. This work has made significant contributions in many aspects. Overall, I believe it is worthy of being accepted.

**Limitations Weaknesses:**

1. I am not an expert in HDL code generation, but the results in table 7 show that the performance of VeriThoughts-14B is not superior to other models. Maybe more HDL code models should be evaluated as the CL-Verilog is just a base model.
2. The performance of Self-Consistent seems not important, table 5 shows that w/ or w/o it achieve better performance on each evaluation dataset. This may limit the contribution of this paper.
3. The overall pipeline is a little bit generating the reasoning thoughts of existing dataset, if the Self-Consistent method is not valid, this feeling is more evident.

**Strengths Contributions:**

1. The reasoning data of  code generation received extensive attention recently, this paper may helps in advancing HDL reasoning.
2. Utilizing the Self-Consistent, the correctness of dataset could be verified.
3. The experiment is comprehensive, almost all the experiment variables are conducted and analyzed.

---

> ### Author Rebuttal · Authors · 2025-07-31
>
> We thank the reviewer for highlighting the benefits our work can bring to code generation for hardware design languages and our comprehensive experiments. We address the limitations and weaknesses brought up by the reviewer below.
>
> **Table 7 results do not show VeriThoughts-14B to have superior performance to other models**: Table 7 showcases our withheld test set “VeriThoughts” and a common Verilog benchmark “VerilogEval”. **We see that VeriThoughts-14B achieves state-of-the-art performance (against open-source models of its size) on the most challenging task for frontier closed-source models, i.e.,  pass@1 on the VerilogEval.** This also demonstrates its ability to generalise to challenging problems from a different distribution outside its training set. VeriThoughts-14B is also second on pass@5 and pass@10 for VerilogEval and on the VeriThoughts withheld test set, being outperformed only by DeepSeek-R1-Distill-Qwen-14B and Qwen3-14B, respectively. These results already suggest that VeriThoughts-14B has superior performance to many other open source code generation models. In addition, we benchmarked the previous state of the art in Verilog generation (RTLCoder) on VerilogEval and showed that VeriThoughts-14B outperforms it.
>
> **Table 5 suggests that the Self-Consistent label is not important**: We discuss the impact of the Self-Consistent label on downstream accuracy in the paragraph started by “RQ2: Does question-answer consistency improve accuracy?” (Line 248). In Table 5, we see that the Self-Consistent label leads to higher accuracy for the majority of the benchmarks, except on the VerilogEval benchmark for the reasoning models. In fact, parallel to our work, two recent papers have also shown that incorrect responses and incorrect reasoning traces can sometimes result in similar or better performance (“Interpretable Traces, Unexpected Outcomes: Investigating the Disconnect in Trace-Based Knowledge Distillation”,”OpenThoughts: Data Recipes for Reasoning Models”) especially when distilling smaller reasoning models, making RQ2 an important research question for the community.
>
> As a “datasets and benchmarking” paper, we believe that the self-consistency labels included in the VeriThoughts dataset are valuable in helping to resolve this open question. In fact, our analysis in Section 3.5 provides some clues as to why this might be the case. Specifically, we find that training data samples with non Self-Consistent labels had longer Verilog code, greater ratio of sequential code, and longer reasoning traces, thus potentially correlating with more challenging problems.  Further, the Self-Consistency labels are also direct evidence of Verilog problems that current frontier models such as DeepSeek-R1 struggle with, and our proposed methodology could be used in building new, challenging datasets for Verilog coding problems.
>
> **Contributions of paper beyond Self-Consistency (SC) labels**:  We note that this paper has several contributions beyond the Self-Consistency labels.   (1) The first Verilog training dataset with associated prompts and reasoning traces for each Verilog module, thus making it possible to distill the first reasoning models for Verilog code generation. (2) An associated data generation pipeline that automatically generates prompts, reasoning traces and SC labels only from any repository of raw Verilog code, making it possible for companies and other entities with large in-house Verilog repositories to replicate our pipeline.  (3) A manual (Table 2) and quantitative (Fig. 3) analysis of SC vs. non-SC datapoints, highlighting the reasons why non-SC data arises (incorrect questions are more common than incorrect responses) and differences in lengths of reasoning traces for two cases.
>  (4) Finally, we perform an assortment of controlled experiments (RQ1, RQ2, RQ3) to showcase the strengths and weaknesses of various subsets of our generated dataset. We believe these contributions significantly elevate the paper above merely generating the reasoning thoughts of an existing Verilog dataset.

---

> > ### Comment · Reviewer_vynn · 2025-08-03
> >
> > Thank you for your reply. It has addressed most of my concerns, so I will increase my score.

---

### Official Review · Reviewer_ZUw5 · 2025-07-03

**Rating:** 5
**Confidence:** 4

**Summary:**

The paper proposes a dataset that focuses on the hardware design language Verilog. Since there is little data to fine-tune Verilog code, the paper generates high-quality samples through LLMs and formal checks. While the LLMs generate the questions, reasoning CoT, and code, the checker verifies whether the generated code is consistent with the groundtruth code. If two codes are consistent, then the generated question is consistent with the groundtruth code. Experimental results validate that the proposed dataset can improve the LLM's performance on Verilog generation.

**Dataset Code Accessibility:**

Yes

**Dataset Code Comments:**

The repository contains detailed instructions and scripts for using the code and data.

**Ethical Considerations:**

No, there are no or only very minor ethics concerns

**Final Justification:**

I have read the authors' response and other reviewers' comments. I have no further concern.

**Limitations Weaknesses:**

- Table 3 lacks the dataset subset "self-consistent+Generated Verilog code". The manual inspection shows that this subset is high-quality and thus should further improve LLM. Why not evaluate this subset and use it for the last VerilogEval experiment?

- In RQ3, the conclusion "hallucinated sample is useful" may result from insufficient high-quality samples. Experiments should remove such data imbalance effect, for example, fine-tune the LLM on VeriGen or MetRex first. This additional training data may also make the comparison with Qwen-3 more fair.

**Strengths Contributions:**

- The paper proposes a reasonable construction process and a novel formal check method. Through manual sample inspection, the paper validates that self-consistency can ensure the correctness of the generated questions and codes.

- The experimental results provide multiple insights into the dataset. They validate the effectiveness of the reasoning CoT, and show that inconsistent samples can also improve the performance.

- LLM trained on the proposed data achieves significant improvements. It also obtains SOTA results on the public benchmarks.

---

> ### Author Rebuttal · Authors · 2025-07-31
>
> We thank the reviewer for highlighting our novel dataset construction method as well as the value of the self-consistency label. We address the limitations and weaknesses brought up by the reviewer below.
>
> **Table 3 Missing Subset**: Since the Generated Verilog code in the Self-Consistent set is already formally verified to be functionally identical to the Ground-Truth Verilog code, we did not additionally train a model on “Self Consistent + Generated Verilog code.” The reviewer makes an excellent suggestion though, it would indeed be interesting to train a model on this subset and compare it with results from the “Self-Consistent + Ground-Truth Verilog code” subset. Since the “Self Consistent + Generated Verilog code” subset is included in the dataset we released, this experiment would be an interesting direction for future research. We will highlight this direction in our paper.
>
> **RQ3 Removing Data Imbalance**: Yes, we agree that this is a very interesting research direction. We believe that by pre-training a model on a large corpus of Verilog data and then fine-tuning it with the VeriThoughts dataset we may see further downstream benefits. We will highlight this direction in our paper.

---

### Note · Authors · 2025-08-14

Thank you to the reviewers for the very thoughtful comments and constructive feedback. We would like to highlight the strengths of our work, as pointed out by the reviewers, and the concerns of the reviewers that we have addressed during the rebuttal stage.

**Strengths**
- The self-consistent label is highlighted as a novel proxy for question quality and helps in the creation of labeled datasets for low resource languages such as Verilog.
- The reasoning traces generated for our dataset provide stronger supervision than traditional SFT and also provide explainability for code generation.
- Our comprehensive experiments show how one can use our dataset to answer various research questions in a systematic manner.

**Reviewer Concerns Addressed**
- Thanks to reviewer comments we have highlighted new research questions and directions such as studying data imbalance in the generated dataset.
- In addition, reviewers have noted the value in studying how our dataset generation framework applies to more varied datasets as well as other languages where formal verification can be used.
- We have clarified the value of the self-consistent label through RQ2 as well as the manual inspection performed in Section 3.5.

---

### Decision · Program_Chairs · 2025-09-18

**Decision:**

Accept (poster)

**Comment:**

**(a) Summary**
The paper introduces *VeriThoughts*, a dataset of 20k Verilog code generation samples with Chain-of-Thought reasoning traces and formal verification labels. It proposes a self-consistency label based on functional equivalence between generated and ground-truth code, enabling high-quality synthetic data curation. The authors also release a benchmark and specialized models, showing improved performance and explainability.

**(b) Strengths**
VeriThoughts is the first dataset to combine reasoning traces with formal verification for Verilog, addressing a critical gap in low-resource hardware design. Its self-consistency labeling ensures data quality, and the release of models and evaluation protocols supports reproducibility. Reviewers praised its novel construction, strong empirical results, and potential to advance trustworthy code generation—fully aligning with the D&B Track’s goals.

**(c) Weaknesses**
Limitations include reliance on general-purpose LLMs for data generation and a lack of evaluation on broader HDL tasks. License metadata was missing on Hugging Face. However, these are minor and do not detract from the core contribution.

**(d) Key Reasons for Acceptance**
The work stands out by integrating **reasoning, formal verification, and dataset generation**—a novel approach for ensuring correctness in code synthesis. The self-consistency mechanism provides a scalable template for other formal domains. By enabling both performance gains and interpretability, VeriThoughts lowers the barrier to research in reliable hardware design, making it a high-impact contribution.

**(e) Rebuttal Summary**
Reviewers questioned the value of self-consistent labels, data imbalance, and generalizability. Authors clarified that inconsistent samples often represent harder problems, and cited recent work showing such data can benefit distillation. They emphasized the pipeline’s extensibility to other Verilog corpora and applications like formal verification. Responses were thoughtful and reinforced the dataset’s research utility.